# Remarkable Structural Modifications of Tialite Solid Solutions Obtained by Different Methods

**DOI:** 10.3390/ma15113981

**Published:** 2022-06-03

**Authors:** Kamil Kornaus, Izabela Czekaj, Natalia Sobuś, Radosław Lach, Agnieszka Gubernat

**Affiliations:** 1Faculty of Materials Science and Ceramics, AGH University of Science and Technology, 30-059 Krakow, Poland; kornaus@agh.edu.pl (K.K.); radoslaw.lach@agh.edu.pl (R.L.); 2Faculty of Chemical Engineering and Technology, Cracow University of Technology, 31-155 Krakow, Poland; izabela.czekaj@pk.edu.pl (I.C.); natalia.sobus@pk.edu.pl (N.S.)

**Keywords:** tialite, DTF simulation, solid solutions, thermal stabilization

## Abstract

The structural changes occurring in tialite due to the formation of magnesium-titanate–aluminum-titanate solid solutions were determined. For this purpose, a DFT simulation of the structural changes was performed. The simulation proposed a number of possible atomic substitutions occurring in the elementary cells of the tialite, along with calculations of the lattice parameter changes in this material. Next, the actual changes occurring in the structure of the tialite due to the formation of solid solutions, obtained in different ways, were investigated. After comparing the obtained results, it was possible to confirm the mechanism of the formation of tialite solid solutions, through which one magnesium atom and one titanium atom substituted two aluminum atoms simultaneously. The results of this experimental work were confirmed by theoretical calculations (the differences in the values of the lattice parameters, measured in the experiment and calculated in the simulation, were less than 0.5%), through which changes in the lattice parameters with Mg and Ti substitution were observed.

## 1. Introduction

Aluminium titanate, or tialite, is a ceramic material used in aluminium metallurgy, mainly due to its low reactivity to aluminium alloys and high thermal shock resistance [1]. The disadvantage of tialite, especially in high-temperature applications, consists in its thermal instability in the temperature range 750–1280 °C, in which it tialite decomposes into its initial oxides, i.e., Al_2_O_3_ and TiO_2_. The specific structure of tialite is the direct cause its thermal instability of [2,3].

Tialite crystallizes in to a pseudobrukite structure. Austin and Schwartz [4] found that tialite is a material isomorphic to pseudobrukite (Fe_2_TiO_5_), with a theoretical density of 3.71 g cm^−3^. Pseudobrukite crystallizes into an orthorhombic arrangement, in the Cmcm space group. Tialite differs slightly in its coordination of its constituent cellular atoms, with lattice parameters a = 3.557 Å, b = 9.436 Å, and c = 9.648 Å. Tialite’s structure consists of distorted octahedra with a central metal cation, surrounded by six oxygen atoms. The unusual combination of octahedra in pseudobrukite-type structures, leading to their deformation, allowed the authors to obtain Me_3_O_5_-type structures instead of Me_3_O_6_-type structures, which are characteristic of octahedra [5]. The TiO_6_ and AlO_6_ octahedra form a network of chains that are weakly connected by their edges in the b and c directions, and by vertices in the a direction. The metal cations forming the deformed octahedra in the pseudobrukite structure tend to occupy positions in their center. This causes their displacement and a decrease in the bond angle O-Me-O to values below 180°, which has a direct effect on the lattice parameter a, whose length is related to the height of the octahedra. Hence, the greater the difference between the ionic radii of the individual metals forming the octahedra, the more they are distorted in the a and c directions [5]. This is particularly evident in the structure of tialite, for which the coefficient of thermal expansion in the a direction takes negative values. A significant deformation of the pseudobrukite structure is the cause of the instability of its compounds. Scala [6] found a relationship between the thermal stability of tialite and its lattice parameter a. The phenomenon, in this case, consists in the decrease in the lattice constant a with the increase in temperature. Thus, the larger the lattice constant a, the higher the thermal stability of the tialite. This constant corresponds to the height of the deformed MeO_6_ octahedra. Increasing its value leads to a decrease in the deformation of the tialite structure. As the elementary cells of tialite are deformed and, therefore, suffers internal stresses, tialite shows thermal instability at temperatures between 750 and 1280 °C. This means that in this temperature range, tialite disintegrates into its constituent oxides, i.e., Al_2_O_3_ and TiO_2_. The introduction of magnesium cations into the structure of tialite leads to the formation of aluminum magnesium titanate (MAT) solid solutions with the structural formula Mg_x_Al_2(1−x)_Ti_1+x_O_5_ [7]. In solid solutions with magnesium, the magnesium cation and the titanium cation simultaneously substitute two aluminum cations; for iron-containing structures, the exchange occurs between Fe^3+^ and Al^3+^ cations [8]. Both mechanisms lead to a reduction in the stresses in the structure, reducing the number and size of micro-cracks, i.e., increasing the thermal stability of tialite. In MAT solid solutions, the elemental cell deformation is lower; thus, the solid solutions exhibit less anisotropy in linear thermal expansion compared to pure tialite. Consequently, lower thermal stresses are generated during their cooling, resulting in fewer and smaller microcracks. Therefore, the obtained materials can be expected to exhibit higher flexural strength while maintaining high thermal shock resistance compared to pure tialite [7].

Tialite is characterized by the high anisotropy of its thermal expansion coefficients [5]. As a result of the formation of a complex stress state in the material, micro and macro cracks are formed in the material during its cooling after reaction sintering [9,10,11,12,13,14,15,16,17,18,19,20,21]. These cracks provide desirable properties, such as a low macroscopic thermal expansion coefficient [22], low thermal conductivity [23,24,25], and, thus, high resistance to thermal shock [1,11,26,27]. Tialite also has high resistance to microcrack propagation [28]. The micro and macro cracks present in the material reduce its flexural strength [20,29,30] and its values for Young’s modulus and Poisson’s number [31]. Therefore, it is necessary to optimize the reactive sintering process and to select stabilizing additives in order to obtain a material characterized by the optimum size and number of micro-cracks, due to which it will have a characteristic resistance to thermal shock while maintaining its mechanical properties.

The synthesis and reaction sintering of tialite solid solutions can be combined by means of the classical solid-phase reaction technique [32,33]. Equimolar mixtures of alumina and titanium oxide are then annealed together with an additive introducing magnesium cations, e.g., MgO or talc [33]. The synthesis using heterogeneous nucleation is a different case in which tialite solid solution is obtained by addition ofisostructural magnesium titanate MgTi_2_O_5_ (MT). This allows the activation energies of the process to be lowered [34,35,36,37], and, thus, the synthesis temperature of the solid solutions to be reduced. In [38], it was shown that reaction sintering leads to an improvement in the homogeneity and density of the sinter. Due to the lower synthesis temperature, the grain size of the obtained sinters is smaller. As a result, finer micro-cracks are formed in the material and its strength is improved [39].

The paper presents changes in lattice parameters of tialite and its solid solutions obtained by reaction sintering. The samples were synthetized with and without stabilizing additives, such as structure-stabilizing additives in the form of magnesium oxide or heterogeneous magnesium titanate nuclei, and with the addition of magnesium cations and silica. The objectives of this work were to investigate whether the method of stabilizer introduction affects the magnitude of the changes occurring in the elementary cells of tialite, whether the addition of SiO_2_ affects the incorporation of Mg^2+^ cations into the tialite’s structure, and to determine the mechanism of incorporation of stabilizer cations into the tialite structure. For this purpose, the results obtained experimentally were compared with the results of the DFT simulations.

## 2. Materials and Methods

### 2.1. Materials

The synthesis of tialite was carried out by means of the solid-state reaction method. TiO_2_ (Reachim, USSR, 99%, d_50_ = 450 nm, rutile) and Al_2_O_3_ (Taimicron, Tokyo, Japan, 99.9%, TM–DAR d_50_ = 120 nm) powders were used for the synthesis. The samples of compositions resulting from stoichiometry of the reaction of tialite synthesis were prepared, as well as the samples into which it was introduced: MgO (Reachim, USSR, 99.5%, d_50_ = 550 nm), MgTi_2_O_5_ (own production, d_50_ = 1.2 µm), and SiO_2_ (Sigma Aldrich, St. Louis, MO, USA, cat no. 381268 99.8%, fumed silica, d_50_ = 6 nm, SBET = 380 g∙m^−2^). The magnesium titanate previously synthesized by means of the solid-state reaction method, with a grain size of about 1 µm, was selected for the study. Table 1 shows the initial compositions used for the study, together with their description used in the text. The addition of 9 and 18% mol. MgTi_2_O_5_ corresponded to 2 and 4% mass. MgO, respectively.

### 2.2. Reaction Sintering of Tialite AT and Solid-Solutions MAT

The systems were wet-homogenized with the use of isopropyl alcohol (POCH, Gliwice, Polska, CAS 67-63-0) and corundum grinders (Cerel, Boguchwała, Polska) (~5 mm). The ratio of powder weight to the grinding media was 5:1. After homogenization, alcohol was removed with the IR irradiator and the powder was granulated in a polyamide sieve. The samples 25.4 mm in diameter and 4.0–5.0 mm in height were formed by double-sided, uniaxial pressing (~8 MPa). Next, the samples were isostatically pressed at 200 MPa. The reactive sintering temperatures of the samples were selected on the basis of DSC measurements [38]. Reactive sintering was carried out at 1250, 1300, and 1350 °C. The temperature increase rate was 5 °C∙min^−1^ and the samples were annealed at the final temperature for 2 h. After the synthesis, the samples were characterized in terms of phase composition (XRD) and microstructure (SEM + EDS).

For the purposes of this work, the synthesis yield data for all synthesized compositions were repeated, according to [38]. On this basis, the samples with high TiAl_2_O_5_-MgTi_2_O_5_ (MAT) solid-solution content and with the most homogeneous, porous-free microstructures were selected for further studies. These compositions included, in particular, those containing isostructural magnesium titanate nuclei: AT 18MT and AT 18MT 4S. The studies of the materials obtained by the novel technique were juxtaposed with the studies of solid solutions, obtained by solid-phase reactions using phases that introduced magnesium cations, in this case, MgO. These included the following compositions: AT 4M and AT 4M 4S. The demonstration of the advantages of MAT solid solutions required the juxtaposition of studies with a reference sample, i.e., a sample without additives, synthesized under the same conditions as previously mentioned. Therefore, the following tests were carried out on the samples: the determination of lattice parameters by XRD method and the observation of microstructure of samples by SEM.

The phase composition of the powders and sinter was determined using an X-ray diffractometer from MalvernPANalytical, Empyrean model (Malvern, Great Britain). The measurements were performed using monochromatic radiation with a wavelength corresponding to the K emission line of copper, in the angular range 5–90° on a 2θ scale, and the step of the goniometer was 0.008°. The qualitative analysis of the phase composition was performed using the X’Pert HighScore Plus version 3.0e computer program developed by PANalytical. To identify the phase composition, the obtained diffractograms were compared with the FIZ Karlsruhe 2012 powder diffraction database and the PDF-2 database (2004). The lattice parameters of tialite and solid solutions were determined using the Rietveld method. First, the theoretical X-ray diffractogram was matched to the real one as closely as possible. Next, the lattice parameters were read from the data sheet of the matched compound.

The microstructure observation and chemical composition analysis were carried out with a Nova NanoSEM 200 FEI Company scanning microscope.

### 2.3. Simulations of the Structure of Tialite and MAT Solid Solutions

In theoretical calculations, the crystal parameters from the literature were used as a starting point [40]. The orthorhombic phase of Al_2_TiO_5_ is described by the space group Cmcm (no. 63), with lattice constants of a = 3.557 Å, b = 9.436 Å, and c = 9.648 Å. The crystal unit cell with alkali metals contained 32 atoms. The metal atoms were located on 4c and 8f symmetry sites and the oxygen atoms on 4c and two 8f symmetry sites [40]. The crystal structure of Al_2_TiO_5_ is shown in Figure 1. The structure of Al_2_TiO_5_ consists of distorted edge-shared oxygen octahedra [MO_6_] surrounding the metal sites, M1 and M2 coordinated to 6 oxygen atoms.

The density functional calculations were performed within the periodic approach of the pseudopotential method, as implemented in the VASP calculations obtained with the projector-augmented wave (PAW) method [41]. The VASP calculations were performed with the gradient-corrected exchange-correlation functional PBE [42]. The sampling of the Brillouin zone was performed using a G-centered k-points mesh generation scheme. The applied k-mesh for the structure optimization was 4 × 4 × 4. At first, the structure optimization was performed on conventional cells of Al_2_TiO_5_, allowing for relaxation of lattice constants and internal degrees of freedom. Subsequently, supercell calculations were performed (exchange of 1 or 2 Al for Ti or Mg atom), in which all degrees of freedom were allowed to relax.

## 3. Results

### 3.1. Phase Composition Analysis by X-ray Diffraction Method

Table 2 summarizes the phase compositions of all the studied samples as a function of the reaction sintering temperature. The analysis of the obtained results shows that for most of the initial compositions, the highest degree of reactivity occurred at the highest reaction sintering temperature of 1350 °C (Table 2). Against this background, two compositions stand out, i.e., the composition with the addition of magnesium cations in the form of MgO (AT 4M), and the composition containing heterogeneous MgTi_2_O_5_ nuclei (AT 18MT). The degree of conversion in these cases was 94–95%. An equally high content of MAT solid solutions for the aforementioned compositions was obtained at 1300 °C, from 89 to 96%. The worst in this comparison was the sample synthesized without any additives, for which the mass % of tialite at 1350 °C was 67.4, and at 1250 °C, according to the DSC analysis presented in [38], no tialite was formed. The synthesis yield of the other studied systems (AT 4M 4S, AT 9MT, and AT 18MT 4S) at the highest reaction sintering temperature (1350 °C) was between 90 and 94%, which can also be considered good results, especially for the seeded samples, i.e., with MgTi_2_O_5_ nuclei. Since both compounds, i.e., tialite (Al_2_TiO_5_) and magnesium titanate (MgTi_2_O_5_), are isostructural, it was impossible to distinguish the reflections from each of them separately. Therefore, it can be concluded that Al_2_TiO_5_-MgTi_2_O_5_ solid solutions, abbreviated as MAT in this paper, were present in the systems with the highest synthesis yield, as well as in the other systems in which magnesium compounds were introduced. In the compositions with the highest synthesis yield, some substrates (1–5 mass %), mainly alumina, were present (Table 2). In the compositions in which magnesium cations in the form of MgO were introduced, trace amounts of spinel were present, while in the samples synthesized with the addition of SiO_2_, as reported by the XRD analysis, mullite was formed by means of the reaction with alumina (Table 2). Therefore, in the samples with the addition of SiO_2_, free TiO_2_ was also identified in the synthesis products (Table 2).

Figure 2 and Figure 3 summarize the changes in lattice parameters for the selected compositions as a function of the reaction sintering temperature. The changes in lattice parameters are illustrated for the reference system, AT, for various systems to which magnesium was introduced in the form of MgO (AT 4M) and MgTi_2_O_5_ nuclei (AT 18MT), and for the analogous systems with nanosilica (AT 4M 4S and AT 18MT 4S), as well as for the nucleated systems as a function of the increased magnesium addition, i.e., AT 9MT and AT 18MT, containing 9 and 18% mol. MgTi_2_O_5_, respectively.

The analysis of the changes in the lattice parameters a, b, and c (Figure 2) shows a significant increase in the values of all the lattice parameters of the samples at each reaction sintering temperature into which magnesium (AT 4M) and magnesium and silica (AT 4M 4S) were introduced compared to the sample synthesized without any additives (AT). Moreover, in the case of the samples with the addition of MgO (AT 4M), the higher the increase in the parameter a, the higher the reaction sintering temperature; this was accompanied by a simultaneous decrease in the values of parameters b and c. In the case of the systems containing both magnesium and nanosilica additives (AT 4M 4S), a smaller increase in parameter a and a smaller decrease in the values of parameters b and c with the increasing reaction sintering temperature (Figure 2) were observed. On the other hand, the analysis of the changes in the lattice parameters, caused by the introduction of magnesium cations in the form of magnesium oxide MgO and the isostructural phase MgTi_2_O_5_, provided further data (Figure 3). The introduction of magnesium cations in the form of magnesium oxide led to a significantly higher increase in the values of the lattice parameters compared to the introduction of magnesium cations via isostructural magnesium titanate nuclei. Additionally, there were noticeably larger changes in all the lattice parameters as a function of the reaction sintering temperature when the magnesium cations were introduced in the form of MgO (Figure 3). Significantly smaller changes in the lattice parameters were observed in the heterogeneously nucleated systems and remained constant from the sintering temperature of 1300 °C onwards.

The changes in the lattice parameters in the samples with increasing concentrations of magnesium cations (AT 9MT—addition of 9% mol. MgTi_2_O_5_; and AT 18MT—addition of 18% mol. MgTi_2_O_5_), summarized in Figure 4, show that higher values of the parameters a, b, and c were observed for the samples containing higher additions of Mg^2+^ (AT 18MT) in comparison with the samples containing fewer magnesium cations (AT 9MT), as well as in comparison with the reference sample, AT.

### 3.2. Results of Structure Simulation Studies of Tialite, Magnesium Titanate, and MAT Solid Solutions

Unit cells with different compositions of Al_2_TiO_5_ were considered. Structure A (Figure 5A) corresponds to the structure of pure tialite. The elementary cells in Figure 5B,C correspond to the situation in which magnesium cations did not build into the tialite structure and the titanium cations partially substituted the aluminum cations. In this case, a non-stoichiometric tialite is formed. Figure 5D shows the structure of pure magnesium titanate. Structures E, F, and G (Figure 5E–G) correspond to the situation in which only Mg_2+_ cations were incorporated into the tialite structure in place of the Al^3+^ cations. This corresponds to the case in which solid solutions described by the formula Mg_x_Al_2(1−x)_TiO_5_ were formed. The last two cases (Figure 5H,I) show examples in which two aluminum cations are substituted simultaneously by one titanium cation and one magnesium cation.

The changes in lattice parameters in the simulations (Table 3, Figure 5) and the experiment (Table 3) were compared with the ideal Al_2_TiO_5_ and MgTi_2_O_5_ structures (Table 4). The specific positions of the atoms in the unit cell are presented in Figure 1.

### 3.3. Thermal Stability

Figure 6 shows the results of the thermal stability measurements of the tialite and MAT solid solutions. The polycrystals with the highest degree of conversion, i.e., reactive-sintered at 1350 °C, were selected for stabilization studies. The samples were annealed for 24, 48, 72, and 96 h at 1100 °C. The thermal stability was determined by the amount of tialite or MAT solid solution after the appropriate annealing time.

The annealing of the AT reference system (Figure 6) at 1100 °C for 24 h led to the almost complete decomposition of the tialite. The initial degree of conversion of the other systems (AT 4M, AT 4M 4S, AT 18MT, and AT 18MT 4S) was high, ranging from 93 to 96 mass % (Table 2). All of these systems showed a significant improvement in thermal stability. For the polycrystals stabilized by the magnesium cations only, the stabilizing effect was greater when the magnesium oxide (AT 4M) was used in comparison to the magnesium titanate (AT 18MT). The complete decomposition of the heterogeneously nucleated materials (AT 18MT) occurred after 3 days of annealing. The complete decomposition of the magnesium oxide stabilized the samples after four days. The most stable system was the one in which the structural (addition of 18 mass % MgTi_2_O_5_) and microstructural (addition of 4 mass % SiO_2_) stabilizations were combined. After 96 h of annealing, the residual solid solution of the MAT was about 40% (Figure 6) in this case. This may indicate that a larger increase in the values of lattice parameters of tialite does not unequivocally lead to an increase in its thermal stability. An equally important role was played by the increased homogeneity of the microstructure and a smaller amount of finer micro-cracks, which may have constituted the nuclei of the decomposition. In addition, the reaction sintering temperatures used made it possible to avoid the excessive grain growth of the obtained tialite solid solutions; consequently, the microstructure of the sinter was characterized by reduced numbers of finer micro-cracks [32]. This may explain the higher thermal stability of the sinters obtained by heterogeneous nucleation, which met the above microstructure criteria, compared with the samples with the addition of MgO. This is despite the fact that the MgO-stabilized samples showed higher increases in the values of their lattice parameters (Figure 3).

## 4. Discussion of Results

The solid-phase synthesis of tialite using oxide substrates can be influenced by many factors. Some of the most important include the synthesis temperature, final temperature annealing time, and substrate grain size. However, controlling only the above synthesis parameters, without the deliberate introduction of magnesium and silica compounds, allows only a 67% yield at 1350 °C. Thermodynamic instability of tialite in the temperature range of 780–1280 °C is an additional parameter worsening the efficiency of the synthesis in the reference systems. It can cause the decomposition of the formed tialite into starting oxides. As was shown in this study, an almost complete conversion of the substrates, of 94–95%, is possible if magnesium cations are introduced into the reaction mixtures in the form of magnesium compounds, i.e., MgO and MgTi_2_O_5_, magnesium titanate isostructural with tialite. For the sample with 4% MgO addition, the synthesis efficiency was already 89% y at the lowest sintering temperature (1250 °C), and from 1300 °C onwards, it was constant at about 96%. Equally high yields were obtained for reaction sintering of the tialite using heterogeneous nucleation. At the lowest temperature, for the compositions AT 9MT and AT 18MT, 85 to 87% of tialite was obtained, at 1300 °C the values ranged from 91 to 93%, and about 95% was obtained at the highest temperatures. Thus, it can be concluded that the presence of magnesium cations effectively activates the synthesis of MAT solid solutions regardless of the form in which they are introduced. The high conversion is particularly important due to the fact that the unreacted grains of the substrate phases in the decomposition temperature range can act as decomposition nuclei, accelerating the decomposition of tialite. Thus, fully reacted systems containing trace amounts of phases other than the MAT solid solution should exhibit much higher thermal stability. The second additive, nanosilica, introduced in an amount of 4 mass %, slightly decreased the amount of the obtained MAT solid solutions. This was probably a result of the formation of the so-called diffusion barriers by silica on the grain boundaries of the crystalline or amorphous phases, hindering the incorporation of Mg^2+^ cations into the tialite structure, i.e., hindering the formation of Mg_x_Al_2(1−x)_Ti_1+x_O_5_ solid solutions. (Table 2). The occurrence of spinels in the systems synthesized using magnesium oxide may indicate that the synthesis of tialite in the systems containing magnesium cations proceeded through the formation of a spinel transition-phase MgAl_2_O_4_. Subsequently, magnesium was incorporated into the tialite structure, and solid solutions of MAT—Mg_x_Al_2(1−x)_Ti_1+x_O_5_ were then formed, in which, according to the literature, one magnesium cation together with one titanium cation probably substituted two aluminum cations simultaneously. Although the inclusion in the stoichiometry of the solid solutions meant that the magnesium cation together with the titanium cation simultaneously probably substituted two aluminum cations in the synthesized products, slight amounts of unreacted oxides, mainly aluminum oxide, were visible (Table 2). This fact can be explained by the higher solubility of the titanium cations in the structures of the resulting solid solutions compared to the aluminum cations. To confirm the above conjecture, an additional experiment was conducted in which compositions were prepared, amounts of alumina that were 2.5, 5, and 10% mol. Less than the stoichiometry of the reaction to form MAT solid solutions. In this way, 100% solid-solution MAT was obtained (Figure 7a). The SEM observations of the sinter made of polycrystal consisting of 100% of MAT solid solution showed that the grain boundaries were very weak, and a significant amount of crumbling of the whole grains of the MAT solution was observed (Figure 7b). This observation suggests that the sintered polycrystal may exhibit low strength. The problem of improving the mechanical parameters of tialite-made polycrystals is now widely discussed in the literature [20,21,27].

The presented analysis of the changes in the lattice parameters of the tialite elemental cell clearly shows that when magnesium is introduced into the systems, both in the form of MgO and MgTi_2_O_5_ alone or together with the addition of nanosilica, MAT solid solutions are formed. A significant increase in the elemental cell volume was observed (Figure 8), in comparison to the elemental cells of the tialite synthesized without additives. The increase in the volume of the elemental cells was larger the more Mg^2+^ cations substituted the aluminum cations, which was in agreement with the reports in the literature. Mg^2+^ (0.078 nm) cations have the largest ionic radius a compared to Ti^4+^ (0.067 nm) and Al^3+^ (0.053 nm) cations, which explains the increase in the elemental cell volume of tialite if substitutions occur.

A careful analysis of Figure 2, Figure 3 and Figure 4, which shows the changes in the lattice parameters (a, b and c), reveals their increase in the systems with the additives compared to the parameters of the elementary tialite cells without any additives (AT). From the point of view of stability, the increase in the value of parameter a is important; due to this increase, the deformation of oxygen octahedra, forming the tialite structure, reduced, the stresses associated with differences in the thermal expansion of the differently oriented grains reduced, fewer micro-cracks formed, and the structure of the MAT solid solutions showed higher stability [1,27,30,33]. It is still worth noting that the increase in the reaction sintering temperature of the tialite or its MAT solid solutions and the associated increase in the synthesis yield resulted in a slight increase in the value of the lattice parameter a, with a slight decrease in the values of the parameters b and c, which can be explained by the formation of the proper structure of the tialite or MAT solid solutions. Furthermore, in the case of the simultaneous application of additives introducing magnesium cations as well as nanosilica additives (AT 4M 4S; AT 9MT 4S and AT 18MT 4S), lower values of lattice parameter a (Figure 2, Figure 3 and Figure 4) and higher values of parameters b and c (Figure 2, Figure 3 and Figure 4) were observed in comparison with the analogous systems not containing nanosilica (AT 4M; AT 9MT and AT 18MT). It can also be seen that the differences in the values of the lattice parameters of the systems with and without silica were negligible when compared to the significant increase in the values of these parameters, relative to the reference system. These observations suggest that silicon does not build into the structure of tialite, while nanosilica forms a kind of “diffusion barrier” on the grain boundaries, limiting the grain size, reducing the number of micro-cracks formed, and decreasing the susceptibility to thermal decomposition, while, on the other hand, hindering the incorporation of Mg^2+^ cations into the structure of aluminum titanate [1,19,27,30,33]. These suggestions were confirmed by the surface phase composition analysis, according to which silicon-rich phases were present at the grain boundaries, at triple points, and between the obtuse grains of the tialite (Figure 9).

The literature data [40,43], used as the basis for the theoretical calculations (Table 3), differ from the obtained results in terms of the values of the lattice constants (structure A—Al_2_TiO_5_ and D—MgTi_2_O_5_ in Table 4). Nevertheless, the crystal structure optimization, using the VASP program, reduced these differences. The lattice constants a and b were smaller by 0.3% and the lattice constant c was larger by 0.24% compared with the values obtained in our experimental studies (Table 5, structure AT). The results obtained for the different Ti and Mg configurations indicate that the Al_2_TiO_5_ structure is highly sensitive to the Al exchange position. The deformation of the elementary cells also changes, depending on the substitution site. A comparison between the differences in the lattice constants in the theoretical modeling and those in the experimental studies indicates that for the exchange of single Al by Ti or Mg, the smallest deviations in the lattice constants, compared to experimental data, were obtained for structures B and E (Table 4 and Table 5). The Ti and Mg positions in structures B and E were used for structure I, whose lattice constants were closest to the experimental data of the AT 18 MT and AT 4M structures. The other structure, H, showed larger deviations, but only up to 0.5%. These results indicate that a configuration in which the exchanged atoms are not in close proximity to each other is preferable. On this basis, as well as on the basis of the literature data, the most probable substitutions are the substitutions of the two aluminum atoms with one magnesium atom and one titanium atom. Moreover, as the simulations show, substitutions are more likely when atoms do not contact each other (Table 4, structure H). The simulations also showed that when the amount of magnesium cation added (AT 9MT) is halved, the values of the real lattice parameters are close to the structures with substitutions of one aluminum cation for one magnesium cation (Table 4, structure E) or one aluminum cation for one titanium cation (Table 4, structure B).

From the presented model (Figure 5H) and its comparison with the real data, it is clear that the current view in the literature [8], starting with the proposition that two aluminum atoms are substituted simultaneously by one magnesium atom and one titanium atom, is highly probable. The elemental cell parameters calculated on this basis are very close to the values obtained from the X-ray data (Table 3). It also seems reasonable to simultaneously apply a structural stabilization with magnesium cations and a microstructural stabilization with silica. Therefore, it is reasonable to believe that the simultaneous substitution of two aluminum atoms with a magnesium atom and a titanium atom increases the volume of the elemental cell and, thus, reduces its distortion, resulting in a less stressed and more stable macroscopic system. These statements are confirmed by the thermal stability studies of the AT tialite and MAT solid solutions. These are illustrated in Figure 6, which shows the amount of tialite or MAT solid solution after an appropriate annealing time.

The almost complete decomposition of the reference sample AT after 24 h of annealing confirms the instability of pure aluminum titanate and indicates the need for stabilizing additives [44,45]. It is reasonable to believe that the lower stability of the AT 18MT and AT 4M4S systems was due to the higher amount of decomposition nuclei, primarily alumina grains and micro-cracks. The sample showing the highest stability of AT 18MT 4S had a highly homogeneous microstructure (Figure 10a), with negligible amounts of unreacted oxide grains and the lowest amount of micro-cracks (Figure 10a). The AT 4M sample, to which magnesium cations in the form of MgO were added and no nanosilica was added, also showed a homogeneous microstructure, although much more micro-cracks were visible (Figure 10b). As a result, after 96 h of annealing, ~40 mass % mass of MAT solid solutions was present (Figure 6). Figure 10c,d shows SEM microstructure images of the AT 4M and AT 18MT 4S samples after 96 h of annealing at 1100 °C. In the case of the sample containing 4 mass % magnesium oxide additive (AT 4M), a clear decomposition and recrystallization of the starting oxides, i.e., TiO_2_ (brightest areas) and Al_2_O_3_ (darkest areas) can be observed. In the AT 18MT 4S sample, on the other hand, the decomposition process was much slower. A few areas can be distinguished in which the recrystallization of the rutile had started (the brightest areas), while the rest of the microstructure consisted of grains of MAT solid solutions, mullite, and spinel [44,45].

## 5. Conclusions

The following conclusions were drawn from the research presented:(1)The values of the lattice constants, obtained from the theoretical calculations, confirmed the experimental data on the changes in the lattice constants under the influence of the substitution of the aluminum centers with magnesium and titanium centers. The differences between the calculated values of the lattice parameters and those determined from the X-ray images were statistically negligible, up to 0.5%.(2)The comparison of the obtained experimental data with the data from theoretical calculations indicates that the most probable mechanism of stabilization of tialite by magnesium cations comprises the simultaneous substitution of two non-adjacent and distant aluminum cations by magnesium and titanium cations (Al_2_TiO_5_ Mg1Ti1 system).(3)An increase in the reaction sintering temperature of tialite leads to an increase in the lattice parameter a in all systems, both for pure tialite and MAT solid solutions, with a simultaneous decrease in the magnitude of the parameters b and c.(4)At the same time, the total volume of the elementary cell does not change significantly. Thus, the stresses in the material at the structural level are reduced, leading to an increase in the thermal stability of tialite. This indicates the significant effect of the reaction sintering temperature of tialite on its stability.(5)The necessity of using stabilizing additives structurally, e.g., Mg^2+^, and microstructurally, e.g., SiO_2_, is indisputable. The presence of magnesium substitutions in the tialite structure significantly reduces elementary cell deformation and the combination of structural and microstructural stabilization by means of nano-silica addition leads to thermally stable polycrystals with desirable properties.(6)The AT 18MT 4S system to which both stabilizing additives, i.e., magnesium cations in the form of isostructured MgTi_2_O_5_ phase and nanosilica, were added showed high thermal stability. After 96 h of annealing at 1100 °C, more than 40mass % of MAT solid solutions were present in the sample.

## Figures and Tables

**Figure 1 materials-15-03981-f001:**
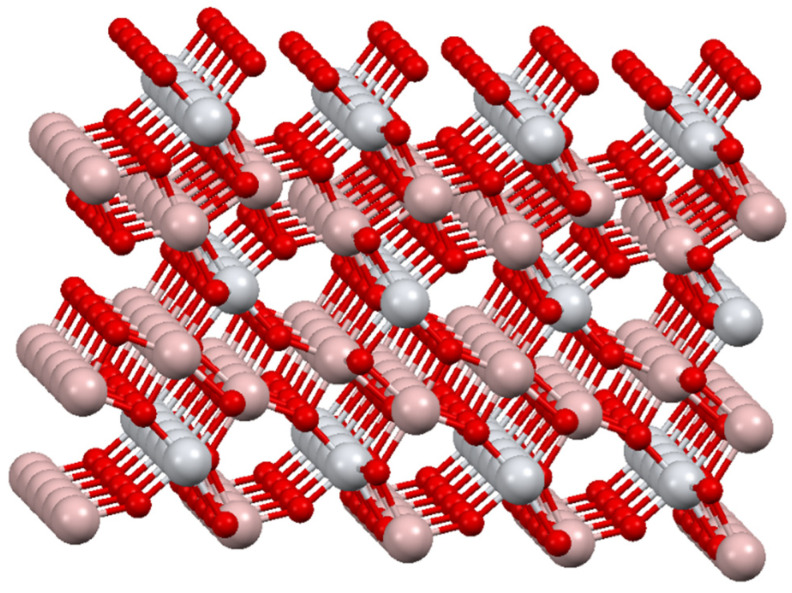
Al_2_TiO_5_ (tialite) bulk structure.

**Figure 2 materials-15-03981-f002:**
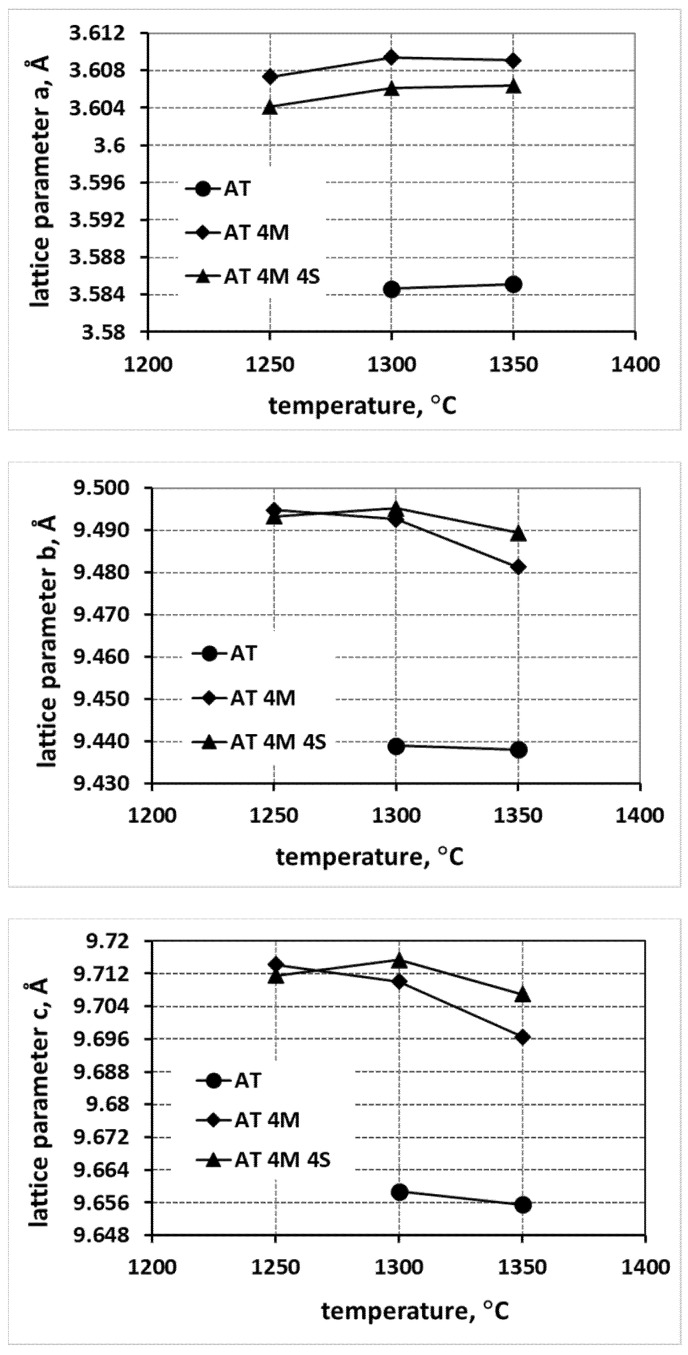
Changes in lattice parameters of tialite elementary cell and its solid solutions in systems stabilized by MgO (AT 4M) and MgO with nanosilica (AT 4M 4S) compared with tialite without stabilizers (AT).

**Figure 3 materials-15-03981-f003:**
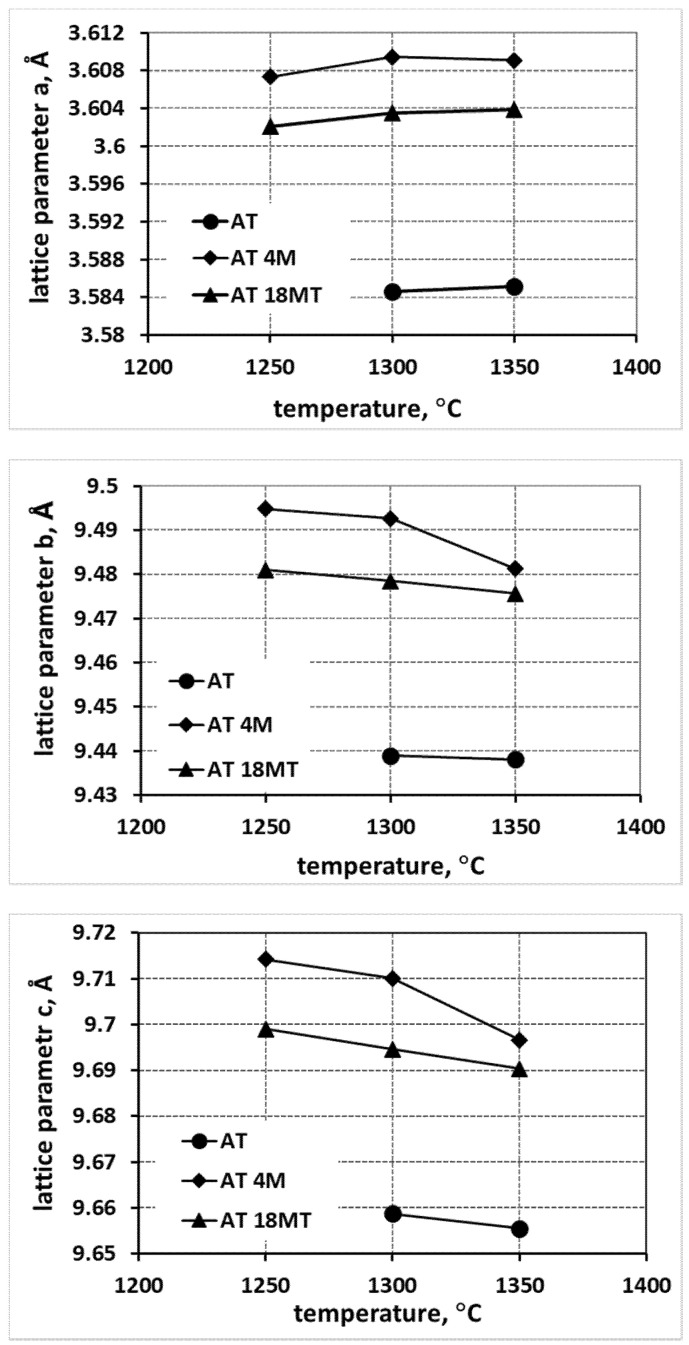
Comparison of changes in lattice parameters of tialite elementary cell and its solid solutions in systems stabilized with the same amounts of MgO (AT 4M) and MgTi_2_O_5_ (AT 18MT) compared with tialite without stabilizers (AT).

**Figure 4 materials-15-03981-f004:**
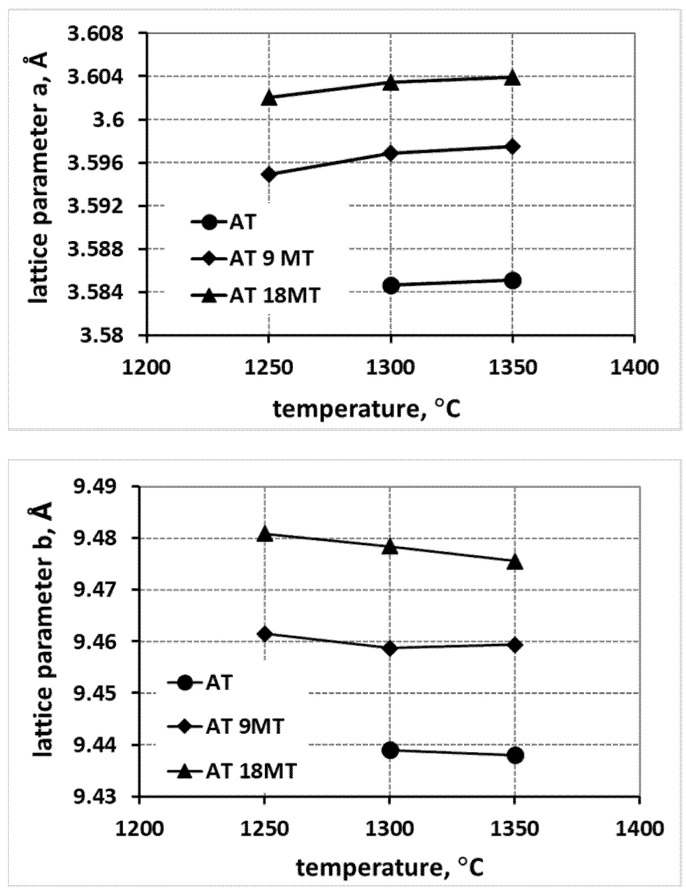
Comparison of changes in lattice parameters of tialite elementary cell and its solid solutions in systems stabilized with different amounts of MgTi_2_O_5_ and tialite without stabilizers (AT).

**Figure 5 materials-15-03981-f005:**
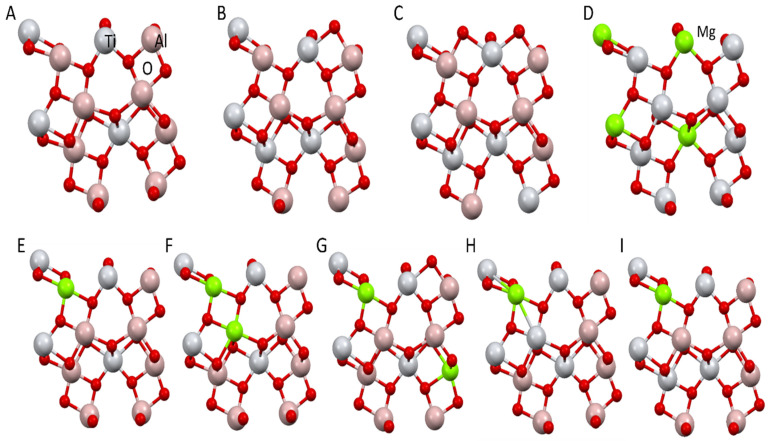
Unit cells with different compositions of Al_2_TiO_5_ (numeration according to Table 3): (**A**) pure calculated by VASP, (**B**) with 1 Al exchanged for Ti, (**C**) with 2 Al exchanged for 2 Ti, (**E**) with 1 Al exchanged for Mg, (**F**,**G**) with 2 Al exchanged for 2 Mg, (**H**,**I**) with 2 Al exchanged for 1 Mg and 1 Ti, and (**D**) MgTi_2_O_5_. 1 Al exchange corresponds to the results for 9% exchange and 2 Al for 18% exchange.

**Figure 6 materials-15-03981-f006:**
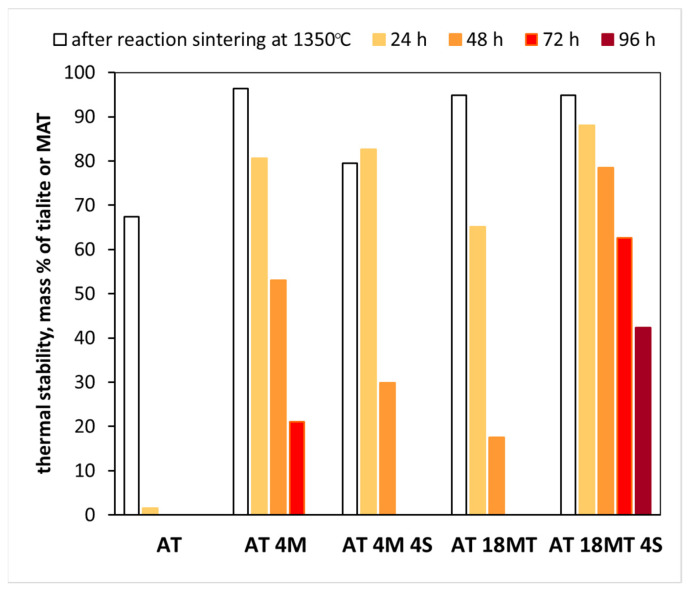
Thermal stability of AT, AT 4M, AT 4M 4S, AT 18MT, and AT 18MT 4S systems after annealing at 1100 °C for 24, 48, 72, and 96 h.

**Figure 7 materials-15-03981-f007:**
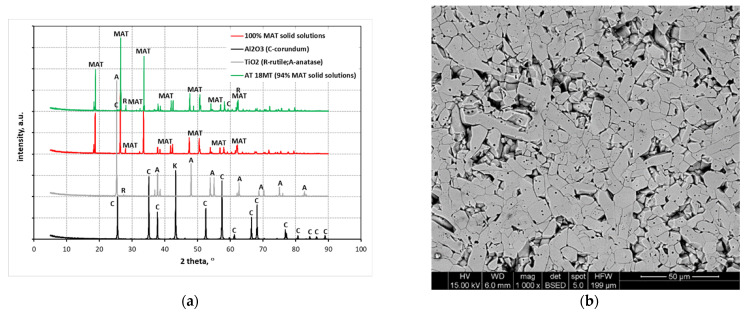
XRD analysis results for a pure MAT solid solution obtained in a non-stoichiometric system (**a**) and SEM image showing the microstructure of a pure MAT solid solution with visible grain chipping formed during polishing (**b**).

**Figure 8 materials-15-03981-f008:**
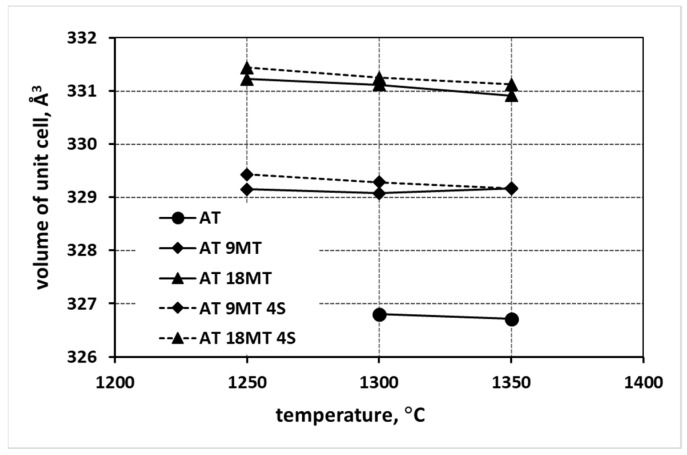
Changes in tialite elementary cell volume as a function of sintering temperature and stabilizing additives used (for comparison, the figure also shows the change in the unit cells of tialite AT).

**Figure 9 materials-15-03981-f009:**
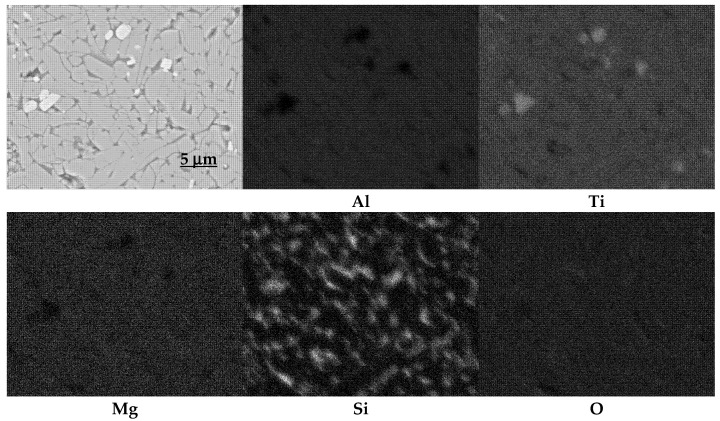
Maps of elemental distribution for sample AT 18MT 4S.

**Figure 10 materials-15-03981-f010:**
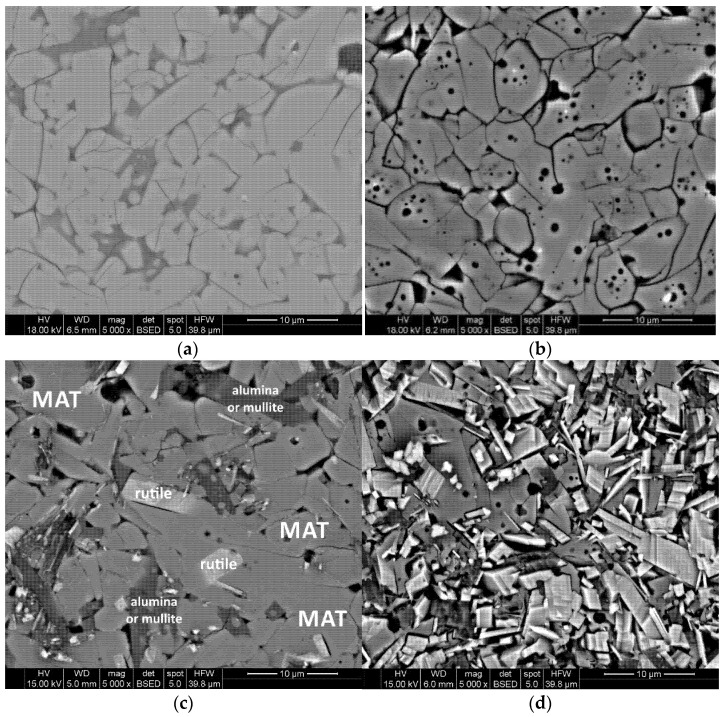
Effect of annealing of MAT solid solutions on the evolution of their microstructure. (**a**) AT 18MT 4S. (**b**) AT 4M. (**c**) AT 18MT 4S_1100 °C/96 h. (**d**) AT 4M_1100 °C/96 h.

**Table 1 materials-15-03981-t001:** Summary of reaction systems used for the investigations.

Reaction System	Symbol
Al_2_O_3_ + TiO_2_	AT
Al_2_O_3_ + TiO_2_ + 4 mass % MgO	AT 4M
Al_2_O_3_ + TiO_2_ + 4 mass % MgO + 4 mass % SiO_2_	AT 4M 4S
Al_2_O_3_ + TiO_2_ + 9% mol. MgTi_2_O_5_	AT 9MT
Al_2_O_3_ + TiO_2_ + 18% mol. MgTi_2_O_5_	AT 18MT
Al_2_O_3_ + TiO_2_ + 18% mol. MgTi_2_O_5_ + 4 mass % SiO_2_	AT 18MT 4S

**Table 2 materials-15-03981-t002:** Phase composition of synthesis products as a function of synthesis temperature and starting substrate composition.

Reaction System	Phase Composition, Mass %																	
	Temperature of Reaction Sintering 1250 °C						Temperature of Reaction Sintering 1300 °C						Temperature of Reaction Sintering 1350 °C					
	**AT**	**MAT**	**K**	**R**	**M**	**S**	**AT**	**MAT**	**K**	**R**	**M**	**S**	**AT**	**MAT**	**K**	**R**	**M**	**S**
**AT**	0	-	59.7	38.9	-	-	62.9	-	29.9	7.2	-	-	67.4	-	26.6	6.0	**-**	**-**
**AT 4M**	-	89.2	5.4	4.3	-	1.1		96.1	2.0	0.9	-	1.0	-	**96.4**	**0.7**	**1.0**	**-**	**1.9**
**AT 4M 4S**	-	73.9	18.1	5.9	2.1	-	-	82.4	3.1	5.1	7.3	2.2	-	**94.4**	-	**2.0**	**3.6**	**-**
**AT 9MT**	-	81.4	16.7	1.9	-	-	-	87.9	11.8	0.3	-	-	-	90.5	9.5	-	-	-
**AT 18MT**	-	86.7	12.7	0.6	-	-	-	91.2	8.8	-	-	-	-	**94.8**	**5.2**	-	-	-
**AT 18MT 4S**	-	82.6	9.8	2.1	3.3	-	-	86.8	4.2	3.5	5.5	-	-	**93.2**	**-**	**1.2**	**5.6**	-

Description of the designations in the table: AT—tialite TiAl_2_O_5_; MAT—solid solutions Mg_x_Al_2(1−x)_Ti_(1+x)_O_5_; R—rutile TiO_2_; K—korundum Al_2_O_3_; M—mullite; S—spinel.

**Table 3 materials-15-03981-t003:** Changes in lattice parameters of Al_2_TiO_5_ after exchange of Al for Mg or Ti, calculated with different amounts of Mg or Ti exchange atoms.

Structure *	Structure	a, Å	b, Å	c, Å
A	Al_2_TiO_5_ vasp	3.572	9.404	9.679
B	Al_2_TiO_5_ Ti1	3.596	9.479	9.678
C	Al_2_TiO_5_ Ti2	3.609	9.568	9.698
D	MgTi_2_O_5_ vasp	3.812	9.608	9.987
E	Al_2_TiO_5_ Mg1	3.594	9.462	9.681
F	Al_2_TiO_5_ Mg1_1	3.631	9.495	9.665
G	Al_2_TiO_5_ Mg2_2	3.625	9.516	9.666
H	Al_2_TiO_5_ Mg1Ti1	3.605	9.514	9.721
I	Al_2_TiO_5_ Mg1Ti1_1	3.630	9.483	9.690

Mg1—1 Al atom exchanged for Mg, Mg2—2 Al atoms exchanged for Mg, Mg1Ti1—2 Al atoms exchanged for 1 Mg and 1 Ti. Exchange of 1 Al atom is equal to around 10% in structure. * Structure according to Figure 5.

**Table 4 materials-15-03981-t004:** Lattice parameters of Al_2_TiO_5_ and MgTi_2_O_5_ based on the literature data [40,43].

Structure	a, Å	b, Å	c, Å
Al_2_TiO_5_ lit.	3.557	9.436	9.648
MgTi_2_O_5_ lit.	3.759	9.824	10.128

**Table 5 materials-15-03981-t005:** Lattice parameters of Al_2_TiO_5_ on the basis of averages of experimental data from Figure 2 and Figure 3 (sintering temperature 1350 °C).

Structure	Structure	a, Å	b, Å	c, Å
AT	Al_2_TiO_5_	3.585	9.433	9.656
AT 4M	Al_2_TiO_5_-MgTi_2_O_5_	**3.609**	**9.481**	**9.696**
AT 4M 4S	Al_2_TiO_5_-MgTi_2_O_5_	**3.606**	**9.489**	**9.707**
AT 9MT	Al_2_TiO_5_-MgTi_2_O_5_	3.597	9.460	9.673
AT 18MT	Al_2_TiO_5_-MgTi_2_O_5_	**3.604**	**9.476**	**9.690**
AT 18MT 4S	Al_2_TiO_5_-MgTi_2_O_5_	**3.602**	**9.480**	**9.670**

## Data Availability

Data will be shared via email upon request from the reader.

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
