# Peer review of "Remarkable Structural Modifications of Tialite Solid Solutions Obtained by Different Methods"

_materials, 2022, doi:10.3390/ma15113981_

Round 1

Reviewer 1 Report

Dear authors, please correct and improve your manuscript according to comments inserted in the manuscript PDF.

Author Response

Dear Reviewer,

Thank you very much for your insightful review. We consider all comments to be valid. We have corrected the editing remarks. We have written explanations for the remaining comments.

Comment 1: “These cracks provide desirable properties of tialite, such as low value of macroscopic thermal expansion coefficient [17], low value of thermal conductivity [18] - [20], low values of Young's modulus and Poisson's number [21], and thus high resistance to thermal shock”

-- check this sentence, please

(the cracks are favorable and the low Young's modulus is desirable?)

Compare this sentence to the one in lines 320-321, as well.

Answer: We agree with the right comment. We have rewritten the above sentence, it now remains in line with the sentence in lines 320-321.

Comment 2: impossible to distinguish the reflections from each of them separately"

--> yet, you have determined the amounts of each of them in Tab 2 (AT and MAT)?

Answer:

The amount of AT tialite was determined only in the system without additives, while MAT solid solutions were determined in systems with the addition of magnesium cations, in the form of MgO or MgTi2O5. Magnesium titanate and tialite are isostructural, so it is impossible to distinguish the different phases, so only the amount of MAT solid solutions (Al2TiO5 - MgTi2O5) was determined in systems with the addition of magnesium cations.

Comment 3: sadly, it seems that the whole manuscript was not proofread from neither of the authors

Odpowiedź: Sorry, we've been reading, but evidently not enough carefully.

Comment 4 : this little sub-chapter (3.2.) does not belong to the chapter "3 Results".

Firstly, it belongs to Materials and Methods.

Secondly, it it too scarcely written, as there is insufficient data on the simulation software that was used.

Answer: The title of subsection 3.2 is misleading; this section contains the results of computer simulations of the structure of tialite, magnesium titanate, and MAT solid solutions. This sub-chapter 3.2 presents results of our calculations shown in Tables 3-4 and compared with summarized experimental data shown in Table 5. We agree that e.g. Figure 5 could be moved to chapter Materials and Methods, however our experience by writing first drafts of this manuscript taught us that it was difficult to follow informations shown in Table 3-4 so if Reviewer kindly agree we would like to keep it in such form to make understanding of changes much full. We included all necessary information about used software (very famous VASP code, which is well known and very good described in literature software) and parameters as in others theoretical publications (including used pseudopotentials, k-points, functional). All those data would be sufficient for experience theoretician to repeat successfully all our calculations using VASP. Our theoretical calculations were made to support experimental data and check if out hypothesis about addition of Mg to tialite are correct, which we successfully proved. Our goal is not to create new type of calculations or methodology, but we are using existing well know methodology implemented in VASP (and well described in literature).

Comment 5: So, a statement from a scientific paper form 1978 is found to be "highly probable" in 2022?

You should reconsider this citation or your statement.

Answer: Recent references mentioning a likely mechanism for stabilization of tialite are from this article. In later years there were publications describing this mechanism as the most probable one, however they referred to the mentioned publication. Our publication is the first to try to prove this thesis.

Comment 6: material strength should not be "guessed" from the image

Answer: Thank you for the right comment, we have softened the statement, we suppose that the material may have low strength since it is destroyed by polishing.

Comment 7: how many measurements (of each  lattice parameter) was done? I guess not one?

if more measurements were done, plotting the confidence intervals on graphs is needed as well

Answer: In polycrystals, phase composition analysis is performed on a set of several thousand crystallites. The determination of lattice parameters is based on the Rietveld method. The nearest X-ray diffractogram from the data sheet is matched to the actual X-ray diffractogram using appropriate equations full of different parameters and constants. Each time an attempt has been made to make the fit as large as possible and thus to make the measurement error as small as possible, also to make the values of elementary cell parameters as precise as possible.

The fit parameters were respectively:

weighted profile R-factor (Rwp) - below 10;

expected R factor (Rexp) - below 10,

as well as GOF (goodness of fit)  - 1-2.

These values indicate a good fit, but their values do not translate into error bars.

Comment 8: so either add (much) more text on describing the simulations that were made, or omit the simulations from the manuscript.

the presented simulation results are too scarce and not presented well.

We moved some information to the results, and deepened the discussion of the results.

The most essential is to understand geometrical/structural changes by addition of Ti or Mg into tialite, which is transparently shown in our opinion only together with theoretical simulations.

Yours sincerely,

Authors

Reviewer 2 Report

Initial comments to the authors:

The authors have used some methods to investigate changes in Tialite structure. The manuscript is well written, but some improvements are required before acceptance. I always worry about the quality of the result presentation and novelty of the work. The study of structural modifications in solid solutions of Tialite is interesting because it can help to improve manufacturing processes for this material. If the authors agree to accept the suggestions below, I will look forward to another round of revisions of this manuscript.

Points of minor concern:

  • Title: please consider changing your title to “Remarkable structural modifications of Tialite solid solutions obtained by different method”.
  • Abstract: The abstract is poor, explicitly, the abstract should contain more quantitative information. Furthermore, every abstract should only contain relevant information that summarizes the work, terms like “then”,... are useless. I recommend authors use the following reference to adjust their abstract (https://doi.org/10.1016/j.carbon.2007.07.009).
  • There are several typographical and grammatical mistakes which should be corrected.
  • A robustly paragraph about previous work on the structure of Tialite should be built on in the introduction. Please consider using the following reference as support (10.1016/j.jmrt.2021.09.112). In addition, other references should be consulted by authors that may help them improve their introduction.
  • The novelty of the work is not expressed explicate. In which aspect this work is original and better than others?
  • Please provide the complete list of chemical and reagents with purity in material and method section using a specific subsection, e.g., 2.1 Materials…2.2. Reaction sintering of MAT solid solutions….
  • The abbreviation used must be explained on their first appearance, or provide separate list of abbreviations.
  • The conclusion of a scientific work, although there may be more than one conclusion, must be constructed in a single paragraph. This paragraph should consist of short, clear and objective sentences so that the reader can quickly understand the written content. Please improve your conclusion section.

Points of major concern:

  • The introduction and result and discussion part should be improved and the results should be interpreted with latest references to make it more understandable for the readers, cite more latest studies related to topic under investigation. A robust comparison of the results presented with others previously published is necessary.
  • How were the lattice parameters obtained from the XRD data? Please give more details of this procedure. Furthermore, I recommend that all patterns obtained be refined in order to deeply study all the structural changes observed due to the use of different methods of obtaining Tialite.

Author Response

Dear Reviewer,

Thank you for performing an insightful review.

As suggested, we have changed the title of the paper. We have removed unnecessary information from the abstract. We have rewritten the introduction and supplemented it with suggested and recent literature. We emphasized the novelty of the work. We completed the characteristics of the chemical reagents. We wrote an appendix in which we defined the abbreviations used in the paper. Instead of a general summary, we wrote the conclusions resulting from the paper. We cannot create a comparison table of the results obtained in our paper and in similar publications because there are no such publications. We rewrote both the introduction and the discussion of the results and enriched them with data from recent publications. In the last question we were asked about lattice parameter measurements based on XRD analysis. We include the answer to this question below.

Question: How were the lattice parameters obtained from the XRD data? Please give more details of this procedure. Furthermore, I recommend that all patterns obtained be refined in order to deeply study all the structural changes observed due to the use of different methods of obtaining Tialite.

Answer: In polycrystals, phase composition analysis is performed on a set of several thousand crystallites. The determination of lattice parameters is based on the Rietveld method. The nearest X-ray diffractogram from the data sheet is matched to the actual X-ray diffractogram using appropriate equations full of different parameters and constants. Each time an attempt has been made to make the fit as large as possible and thus to make the measurement error as small as possible, also to make the values of elementary cell parameters as precise as possible.

The fit parameters were respectively:

weighted profile R-factor (Rwp) - below 10;

expected R factor (Rexp) - below 10,

as well as GOF (goodness of fit)  - 1-2.

These values indicate a good fit, but their values do not translate into error bars.

Best regards,

Authors

Reviewer 3 Report

Based on the previous recommendation, the details about discussion have been added and revised. The quality of revised work is significantly improved. All the issues raised by the reviewers have been addressed well. Therefore,it can be accepted for publication in current version.

Round 2

Reviewer 1 Report

Dear authors,
Thank you very much for your extensive efforts invested in correcting and improving the manuscript. I am glad I was able to help. Best regards

Reviewer 2 Report

This manuscript can be accepted for publication.